# Features of Industrial Green Technology Innovation in the Yangtze River Economic Belt of China Based on Spatial Correlation Network

**Mengchao Yao [1], Ziqi Li [2,*] and Yunfei Wang [1,*]**

1  Business School, Soochow University, Suzhou 215006, China
2  School of Economics & Management, Huzhou University, Huzhou 313000, China
*  Correspondence: qizili1993@163.com (Z.L.); 20204010009@stu.suda.edu.cn (Y.W.)

**Abstract:** A generic phrase for technical and managerial innovation geared toward environmental conservation is "green-technology innovation." It is essential to attain ecologically friendly development that promotes economic progress. Promoting the combined growth of the economy, society, and environment is extremely important. The industrial-green-technology innovation efficiency of 110 cities in the Yangtze River Economic Belt is calculated using the Sup-SBM model from 2011 to 2021 while considering undesirable output. The modified gravity model is then used to convert the attribute data of industrial-green-technology innovation efficiency into relational data. The Yangtze River Economic Belt uses the social-network-analysis (SNA) approach to investigate the geographical correlation-network properties of industrial-green-technology innovation efficiency. The findings demonstrate the following: (1) There is a rising trend in the degree of industrial-green-technology innovation efficiency between different cities in the Yangtze River Economic Belt, and this pattern is known as "three plates." (2) The examination of network characteristics reveals an indigenous core–edge structure in space, with the network density of the Yangtze River Economic Belt displaying an increasing trend over the research period. (3) Individual characteristic analysis reveals that although the innovation-efficiency network tends to be flat, the degree centrality and closeness centrality of industrial-green-technology innovation efficiency in the Yangtze River Economic Belt indicate an upward trend over the research period. In addition, Chengdu in the upstream region, Wuhan in the center, and Shanghai in the downstream area serve as bridge and intermediary nodes in the spatial correlation network. (4) Block-model analysis reveals a close spatial link between blocks. A more complex and durable spatial link is now possible because of the spatial relationship of green-innovation efficiency in cities, which has shattered the boundaries imposed by traditional geographic space. The Yangtze River Economic Belt will be jointly promoted by several of the policy recommendations in this paper, aligning with that.

**Keywords:** industrial-green-technology innovation; modified gravity model; social-network analysis; space-related networks

## 1. Introduction

Due to China's dual characteristics of innovation-driven and green development, green-technology innovation has emerged as a vital driver of the country's high-quality economic growth in the conflict between economic development and the issue of resources and the environment [1]. According to the 14th Five-Year Plan, green is the most recognizable backdrop for China's development. Implementing a green economic policy is necessary for the new standard development stage. Innovation in green technology is quickly emerging as a critical area of the most recent industrial revolution and international rivalry in science and technology. It is now a crucial component of efforts to combat pollution, build an ecological civilization, and advance high-quality development. The historical stages of industrialization and urbanization are continuously being experienced in

China. There are still a lot of conventional industries. High-tech and strategically important developing industries are still not the main drivers of economic growth. The low energy efficiency and coal-biased energy structure have stayed the same. Practical solutions to pollution issues in crucial areas and important businesses have yet to be found [2]. Resources and the environment are more severely limited. There is a small window between the carbon peak and carbon neutralization. There needs to be more than the available technical resources. Promoting the green and low-carbon transformation of industry is complicated. Consequently, the importance of green-technology innovation in the process of the green development of China's industry is rising, and it has turned into a crucial tool for industrial enterprises to deal with the crisis of resource consumption, find a solution to the issue of environmental pollution, and realize green transformation and development [3]. The report of the Communist Party of China's 19th National Congress proposed the development of a system for market-driven green-technology innovation that further supports the requirement and strategic importance of using industrial enterprises as the primary engine of green-technology innovation [4]. The eastern, central, and western regions are all included in the Yangtze River Economic Belt. The endowment of resources, the stage of economic growth, and the issues and development priorities of ecological-environment protection are all significantly different among regions. The spatial-correlation impact of the regional economy is growing under the influence of mobile space at the same time as geographical spatial restrictions are gradually loosening with the establishment of a regional coordinated development strategy. The network paradigm is replacing urban-relations theory [5]. To effectively leverage regional comparative advantages, advance ecological-environment protection and economic development, and increase the high-quality, coordinated development mechanism of the Yangtze River Economic Belt, it is crucial to investigate the spatial-correlation structure and evolution trend of industrial-green-technology innovation in the Yangtze River Economic Belt under the new development concept.

Green-technology innovation has positive and negative externalities based on knowledge, technological spillover, and the public attribute of natural resources [6]. The first studies on green-technology innovation mainly focused on its meaning and traits. The term "green technology" was first introduced by Brawn and Wield [7] in 1994. Since then, they have carefully explained its evolutionary stage, emphasized its key components, and merged previous ecologically related technologies. They believed "green technology" was a catch-all phrase for techniques and procedures, including lowering energy and environmental pollution, highlighting its importance for sustainable economic and social development, and upgrading in response to green growth demands. To reduce the negative externalities on the ecological environment in the production and consumption processes, or "green technological innovation," academics are gradually examining the life cycle of green production processes and green ecological products [8]. The decoupling of economic growth from resource consumption and the tunneling of the environmental Kuznets curve are two benefits of green-technology innovation that are realized when taking into account the goal of sustainable development [9]. These benefits are realized through the synergy of development goals, the diversification of innovation subjects, and the sharing of achievement benefits for the public welfare. To provide a more focused experience for developing pertinent national policies, scholars increasingly move their attention from theoretical to empirical analysis as their research becomes more in depth. The evaluation of green-technology innovation efficiency [10,11], motivators [12,13], the geographical effect [14,15], and other features are all steadily covered in a related study. In assessing the effectiveness of green-technology innovation, two standard methods are the parameter technique, based on stochastic frontier analysis (SFA), and the non-parameter method, based on data-envelopment analysis (DEA) [16]. The data-envelopment analysis method is more prevalent among academics since it can easily handle numerous input and multiple output indications and does not require particular production-function forms to be established. As areas become more interconnected, experts' focus has turned to the spillover of green-technology dissemination, and spatial considerations have steadily been

added to the analysis framework of influencing factors. Currently, researchers mostly employ the Gini coefficient [17], spatial exploratory-data analysis [18], kernel-density function [19], and spatial convergence [20] to quantify and assess the factors that drive the development of green technologies at various spatial scales. The majority of current studies, however, study the geographical-pattern aspects of green-technology innovation efficiency using attribute data [21]. In addition to various local attributes, the level of innovation in green technology in a given area is also influenced by the relationship between the intervals [22]. However, most current studies based on new economic geography and spatial econometric models concentrate on the proximity spatial effect and local spatial characteristics when examining the impact of interval relationships. These studies also need more systematic analysis of the overall network association and reveal the overall spatial pattern and driving mechanism of green-technology innovation efficiency. To better support the rational allocation of innovation resources, realize the overall optimization of the green-industrial-technology innovation-efficiency spatial pattern, and provide scientific reference for cross-regional collaborative promotion, this paper analyzes the evolution characteristics and operating mechanism of the spatial-correlation-network structure of green-technology innovation efficiency based on relational data from the perspective of a complex network.

The contributions of this paper are as follows: (1) Cities are the most basic unit of economic development [23] and urban innovation is the basis of regional innovation [1]. This paper takes 110 prefecture-level cities in the Yangtze River Economic Belt as the research object. It uses the Super-SBM model, considering undesirable outputs, to measure the efficiency of industrial-green-technology innovation. (2) From the perspective of spatial correlation, the gravity model is modified [24] and the social-network analysis method is used to analyze the network structure of industrial-green-technology innovation efficiency in the Yangtze River Economic Belt. It is of practical significance to clarify the relationship between the self-development of various regions in the Yangtze River Economic Belt and the coordinated development of the whole basin regarding green-technology innovation efficiency and to reveal the formation mechanism of a spatial-correlation network.

## 2. Research Methods and Data Sources

### 2.1. Efficiency Measurement of Industrial-Green-Technology Innovation—Super-SBM Model

2.1.1. Measurement-Model Selection

The present research focused on using non-parametric techniques based on data-envelopment analysis (DEA) to gauge and assess the effectiveness of green innovation [2]. The two primary approaches to data-envelopment analysis are the conventional DEA model and the highly efficient SBM (slacks-based measure) model. However, the former disregards the relaxation of input or output, which can cause the efficiency measurement to deviate and does not support the correctness of the assessment results [25]. The latter adds unexpected output indicators, which not only overcome the inaccuracy brought on by a single output indicator but also solve the issue of high efficiency caused by the former without taking into account slack variables, making the evaluation results more scientific and reasonable [26], which can be used as an essential reference for the efficiency measurement of industrial green innovation. This research utilized the Super-SBM model with unexpected output to assess the effectiveness of the Yangtze River Economic Belt's industrial green innovation. The model is described as follows: Assuming 110 decision-making units, each decision-making unit contains three vectors: input $X$, expected output $Y^g$, and unexpected output $Y^b$. If m units of input are consumed, s1 and s2 units of expected output and unexpected output will be generated, respectively. Therefore, these

three vectors are expressed as $x \in R^m$, $y^g \in R^{s_1}$, and $y^b \in R^{s_2}$ respectively, and set to matrix $X$, $Y^g$, and $Y^b$, respectively, as follows:

$$
\begin{aligned}
X &= [x_1, x_2 \cdots x_n] \in R^{m \times n} \\
Y^g &= \left[y_1^g, y_2^g \cdots y_n^g\right] \in R^{s_1 \times n} \\
Y^b &= \left[y_1^b, y_2^b \cdots y_n^b\right] \in R^{s_2 \times n}
\end{aligned}
\tag{1}
$$

In Equation (1), $x_i > 0, y_i^g > 0, y_i^b > 0 (i = 1, 2 \cdots n)$. The Super-SBM model based on unexpected output can be expressed as:

$$
\rho^* = \min \rho = \frac{1 - \frac{1}{m}\sum\limits_{i=1}^{m}\frac{S_i^-}{x_{i0}}}{1 + \frac{1}{s_1+s_2}\left(\sum\limits_{r=1}^{s_1}\frac{S_r^g}{y_{r0}^g} + \sum\limits_{r=1}^{s_2}\frac{S_r^b}{y_{r0}^b}\right)}
$$
$$
S.t. \begin{cases} x_0 = X\lambda + S^- \\ y_0^g = Y^g\lambda - S^g \\ y_0^b = Y^b\lambda + S^b \\ S^- \geq 0, S^g \geq 0, S^b \geq 0, \lambda \geq 0 \end{cases}
\tag{2}
$$

In Equation (2) $S^-, S^g, S^b$ represents the slack variables of input, expected output, and unexpected output, respectively. $\lambda$ is the weight vector. The objective function $\rho*$ represents the efficiency value of the evaluated decision-making units (DMUs) and strictly monotonically decreases for $S^-, S^g, S^b$. The larger $\rho*$ is, the more efficient the decision-making unit is.

### 2.1.2. The Evaluation-Index System Construction

Since the efficiency of green-technology innovation is different from the traditional technology innovation efficiency that pursues economic interests and is different from the input–output efficiency of environmental technology innovation, financial, environmental, and social benefits should be unified. Therefore, this paper referred to previous studies [8,27,28] to construct the industrial-green-technology innovation-efficiency input–output-index system in the Yangtze River Economic Belt, as shown in Table 1.

**Table 1.** Input–output–index system of industrial green-technology innovation efficiency in the Yangtze River Economic Belt.

| First-Level Indicators | Second-Level Indicators | Third-Level Indicators | Basic Index | Unit |
|---|---|---|---|---|
| The efficiency of industrial-green-technology innovation | Input index | R&D personnel input | The full-time equivalent of R&D personnel in industrial enterprises above scale | Person/year |
| | | R&D fund input | Internal R&D expenditure of industrial enterprises above scale | CNY 10,000 |
| | | Industrial energy input | Industrial energy consumption | 10,000 tons of standard coal |
| | Expected-output indicators | Number of patent applications | Number of effective invention patents for industrial enterprises above scale | Item |
| | | Sales revenue of new product | New product sales income of industrial enterprises above scale | CNY 10,000 |
| | Unexpected-output indicators | Industrial wastewater | Total industrial wastewater discharge | 10,000 tons |
| | | Industrial waste gas | Total industrial SO$_2$ emission | 100 million cubic meters |
| | | Industrial solid waste | Output of general industrial solid waste | 10,000 tons |

Energy, capital, and workforce are examples of input indicators. This study compared energy factors to traditional innovation components to determine how well the Yangtze River Economic Belt produces industrial green innovations. The R&D funding input adopted internal R&D expenditure in companies above size, and the R&D personnel input

chose a full-time equivalent of R&D workers in industrial enterprises above scale based on data availability. The industrial-energy input selected 10,000 tons of standard coal as an alternate metric for overall industrial energy consumption.

The expected output and the anticipated output of fees are included in the output indicators. The number of patent applications and the sales revenue of new items makes up the predicted output. The former is indicated by the number of industrial firms with active invention patents larger than the specified size. The latter is represented by the new product sales revenue from industrial companies more significant than the chosen size. The number of patent applications and the sales revenue of new items makes up the anticipated output. The former is indicated by the number of industrial firms with active invention patents larger than the specified size. The latter is represented by the new product sales revenue from industrial companies more significant than the selected size. In selecting unexpected-output indicators, the total discharge of industrial wastewater and general industrial solid waste was used to characterize industrial and industrial solid waste, respectively. Considering data availability, industrial sulfur dioxide ($SO_2$) emissions replaced industrial-waste gas emissions.

### 2.2. Correlation-Measure Method—Modified Spatial-Gravity Model

The study of actor relationships and quantitative analysis of diverse interactions can be accomplished through social-network analysis. As a result, before performing network analysis, the connection must be established [29]. The two most popular techniques for determining actors' relations are the gravity model and the Granger causality test [30]. The gravity model thoroughly considers the influence of economic and geographical distance and scientifically and effectively guarantees the accuracy of spatial correlation, making it more appropriate for analyzing all of the cross-sectional data [31]. The VAR Granger causality model has an extreme time-lag sensitivity and does not apply to data within a short period. Hence, the modified gravity model describes the Yangtze River Economic Belt's 110 cities' spatial-correlation strength.

$$R_{ij} = K_{ij} \times \frac{E_i \times E_j}{D_{ij}^2 / (g_i - g_j)^2}$$
$$K_{ij} = \frac{E_i}{E_i + E_j} \tag{3}$$

In Equation (3), $R_{ij}$ represents the correlation strength of industrial-green-technology innovation efficiency between the two cities. $K_{ij}$ is the gravitational coefficient, and $E_i, E_j$ is the industrial-green-technology innovation-efficiency value of the city $i, j$. $D_{ij}$ is the geographical distance between cities $i, j$, $g_i, g_j$ is the per capita GDP of the city $i, j$, and $g_i - g_j$ is the economic distance between cities $i, j$.

The Yangtze River Economic Belt's industrial-green-technology innovation-efficiency gravity matrix can be obtained using the gravity model. The crucial value is determined by taking the average data value for each row in the matrix. A spatial link between the city's industrial-green-technology innovation efficiency and the city in the column is shown by the data value higher than the critical value in the same row, which is 1 in this case. The gravity matrix between industrial-green-technology innovation becomes relationship data when the value is 0, creating a directed 1-model network that serves as the database for the social-network analysis method [32]. In contrast, 0 denotes a lack of spatial correlation between cities. The intensity of the town's association is set to 0 to prevent a closed subring.

### 2.3. Characteristic Measurement Method of Complex Relationship—Social-Network Analysis

The point in the spatial-correlation network and the line connecting the point in the network are the primary subjects of the social-network analysis approach. The grid's degree, density, efficiency, and grade indicate the whole network's qualities. The different network features are described using the three centrality indices. The location relationship in the spatial network from clustering is studied using block-model analysis. Therefore, social-network analysis can reveal the uniqueness, hierarchy, and integrity of its spatial

network, as well as investigate the relationship between the tightness of the spatial corre­lation and network nodes and comprehensively reflect the spatial-network structure and characteristics of industrial-green-technology innovation efficiency in the Yangtze River Economic Belt [33].

2.3.1. Overall Network Characteristics

The overall network structure is composed of all members within the group and their relationships, which can reflect the relationship between the strength, grade, and stability of the entire network [34]. This paper uses correlation indexes such as network density, network correlation degree, network efficiency, and network hierarchy to describe the spatial-network structure of industrial-green-technology innovation efficiency in the Yangtze River Economic Belt.

The network-density index reflects the degree of closeness of the relationship between regions in the network. The ratio of the actual correlation number to the theoretical maximum possible correlation number in the network calculates the network density. The expression of network density is as follows:

$$C_D = \frac{L}{N(N-1)} \tag{4}$$

$L$ represents the number of relationships in the network and $N$ represents the number of nodes. In this paper, $N = 110$, $N(N-1)$ represents the maximum possible value of the total number of relationships in the overall network in theory. The value range of network density $C_D$ is [0, 1]. The closer the network density is to 1, the closer the correlation between nodes is. On the contrary, the sparser the correlation between nodes is.

Network correlation is an index to measure the accessibility between nodes in complex networks, which is used to characterize the robustness and vulnerability of spatial networks. Suppose each node in the spatial-correlation network can be directly or indirectly connected through other points. In that case, the network correlation-degree value is 1, indicating that the network correlation is high and any node in the network can reach another node. The expression of network-correlation degree is as follows:

$$C_N = 1 - \frac{V}{N(N-1)/2} \tag{5}$$

$V$ represents the number of points that are not accessible to a directed network.

Network efficiency refers to how many redundant lines exist in the network under the condition that the complex network contains the number of nodes—that is, to measure the connection efficiency between each point in the network. The lower the network efficiency, the more multiple lines there are in the spatial correlation network—that is, the higher the relationship transmission path between the two nodes, the stronger the network stability; conversely, it shows that with fewer redundant paths from one node to another in the network there is a lack of effective collaboration between node cities. The expression of network efficiency is as follows:

$$C_E = 1 - \frac{S}{\max(S)} \tag{6}$$

$S$ represents the actual redundant lines in the network and $\max(S)$ represents the possible maximum redundant lines.

The network-hierarchy index measures the degree of asymmetric reachability between nodes in the network, reflecting the dominance or hierarchy of nodes in the network. The higher the network level is, the more pronounced the core–edge structure is, and vice

versa—the role or status of each node city in the network is similar. The expression of network hierarchy is as follows:

$$C_H = 1 - \frac{M}{\max(M)} \tag{7}$$

$M$ is the symmetric reachable point number in the network and $\max(M)$ is the maximum possible symmetric reachable point number.

### 2.3.2. Individual Network Characteristics

The individual network structure is composed of an individual and other individuals directly connected to it, which can reflect the position and role of individuals in the spatial-correlation network [35]. This paper uses centrality indicators such as degree centrality, closeness centrality, and intermediate centrality to characterizing the individual network-structure characteristics of industrial-green-technology innovation efficiency in the Yangtze River Economic Belt.

Degree centrality, which reflects the city's position in the spatial-correlation network, is the number of other nodes directly related to the node. The more central a town is in the network, and the more spatially correlated its industrial-green-innovation efficiency is with other cities, the higher the degree of centrality is [36]. The degree of each point in the directed graph can be further broken down into in-degree centrality and out-degree centrality, which, respectively, reflect the direct spillover from other regions and the level of that region's spillover to other areas of the network for green-technology innovation [37].

The proximity centrality reflects how a region does not control other regions. The higher the proximity centrality is, the shorter the distance between one region and other regions is and the more significant the direct correlation between industrial-green-technology innovation efficiency in the city and other cities is [38]. The expression of closeness to centrality is as follows:

$$C_{APi}^{-1} = \sum_{j=1}^{N} d_{ij} \tag{8}$$

$d_{ij}$ represents the shortcut distance between city $i$ and city $j$.

Intermediary centrality describes the ability of node cities to control information and resources in the network. The higher the intermediary centrality is, the stronger the power of node cities to prevent other node cities is and the more prominent the role of the intermediary and bridge is [39]. The lower the intermediate centrality of the node, the weaker the control ability of the node to other nodes, and the node is at the edge of the whole network. The expression of intermediary centrality is as follows:

$$C_{RBi} = \frac{2\sum\limits_{j}^{N}\sum\limits_{k}^{N} b_{jk}(i)}{N^2 - 3N + 2}, (j \neq k \neq i, j < k) \tag{9}$$

$b_{jk}(i) = g_{jk}(i)/g_{jk}$ denotes the probability of city $i$ in the shortcut between city $j$ and city $k$, $g_{jk}$ represents the number of shortcuts between city $j$ and city $k$, and $g_{jl}(i)$ denotes the number of shortcuts between city $j$ and city $k$ passing through the third city $i$.

### 2.3.3. Block Models

Block-model analysis, which White and Harrison first presented in 1976, is a crucial tool for social-network spatial-clustering analysis [40]. Block-model analysis can investigate the network's internal structure and the placement and function of different sections within the plate, specifically looking at information-transmission and -reception methods between different scales and conducting descriptive analysis. The role of each plate is primarily divided into four plates based on the number of sending and receiving relations of cities in each plate [41]: The first is the two-way spillover plate, which sends and receives influence

from other vessels and has a large number of connections among its members; the second is the net-overflow plate, where the number of relations issued by the container to other dishes is greater than the number of ties received from other dishes; the third is the net-benefit plate, which receives more relations from other plates than it sends out; and the fourth is the agent plate, which has fewer relationships among members within the plate but more links with other plate members.

The specific division method is as follows: Assuming that there are n economic entities in the whole network, there is no difference in each economic entity, and there are k members in plate i, then the number of relations that can occur between plate i and the whole network is $k(n-1)$ and the number of relations that can occur within plate i is $k(k-1)$. It can be concluded that the expected internal-relationship ratio of plate i to the whole is $k(k-1)/k(n-1) = (k-1)/(n-1)$. Let the internal relationship between the members of plate i be $V_i$ and the actual relationship between the members of plate i in the whole network be $V_{i,n}$. The outflow relationship of plate i to other plates is $V_{i,n-k}$. The inflow relationship of plate i to other plates is $V_{n-k,i}$. The block-model division of the multi-valued network is shown in Table 2.

**Table 2.** Multi-valued network block-model partition.

| Relationship between the Various Sections | Net Inflow/Net Outflow Relationship between Block I and Other Blocks | |
|---|---|---|
| | $V_{i,n-k}/V_{n-k,i} >= 1$ | $V_{i,n-k}/V_{n-k,i} < 1$ |
| $V_i/V_{i,n} >= (k-1)/(n-1)$ | Bi overflow | Net benefit |
| $V_i/V_{i,n} < (k-1)/(n-1)$ | Net overflow | Agent |

*2.4. Data Sources*

Taking 110 prefecture-level cities in the Yangtze River Economic Belt as the research object, considering the availability of data, this paper took 2011–2021 as the research period. Because of the time lag between the input and output indicators, the time lag was set to 1 year, the input index was 2010–2020 data, and the output index was 2011–2021. The patent authorization in the output index was expected to come from the patent-retrieval system of the State Intellectual Property Office of the People's Republic of China. Other data were mainly derived from the China Statistical Yearbook, China Environment Statistical Yearbook, China Energy Statistical Yearbook, China Science and Technology Statistical Yearbook, and the statistical bulletin of cities. The missing data were obtained by interpolation and extrapolation. The geographical distance between cities is the spherical distance calculated according to the latitude and longitude of the two cities.

## 3. Result Analysis

*3.1. Spatial-Association Strength*

Since the gravity between the two cities is directional, the sum of the gravity values between the two cities was used as the connection strength in the calculation results when using gravity model (3) to calculate the gravity value of industrial-green-technology innovation efficiency in the Yangtze River Economic Belt. The Yangtze River Economic Belt's 110 prefecture-level cities served as the paper's research subject. The theory provided 5990 link strengths between cities and 11,990 pairs of gravity values. This research chose cities with the top 5% of spatial-correlation strength in 2011 and 2021 as the analysis samples to more intuitively characterize the general association of cities in the Yangtze River Economic Belt. Four categories—strong link, medium-strong connection, medium-weak connection, and weak connection—were used to categorize the correlation strength. Visual displays were created using ArcGIS software, as seen in Figure 1.

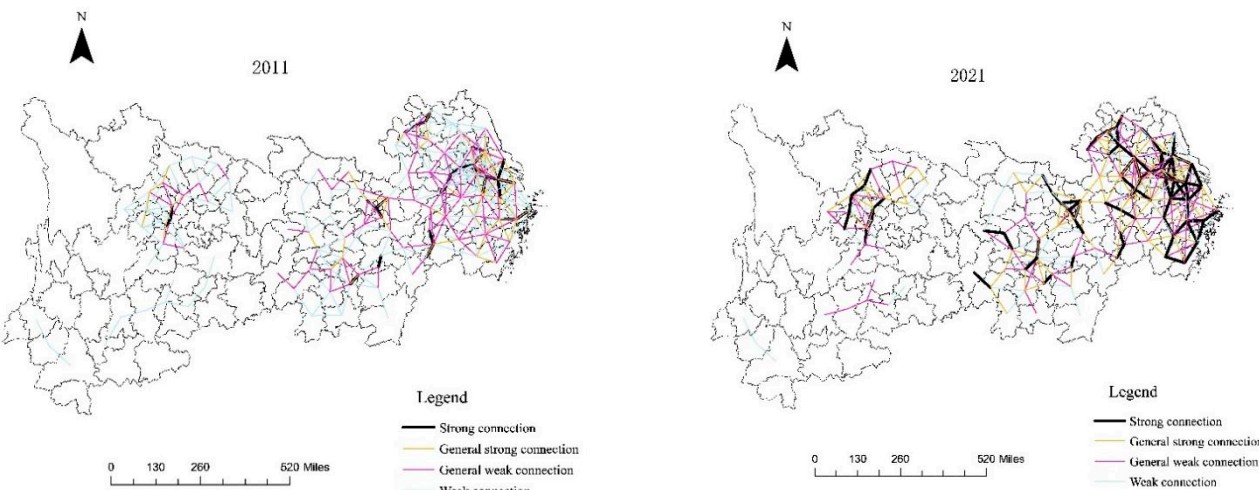

**Figure 1.** Top 5% correlation strength of industrial-green-technology Innovation Efficiency in the Yangtze River Economic Belt.

　　Generally, there was a growing tendency in the degree of green-industrial-technology innovation-efficiency linkage between various cities in the Yangtze River Economic Belt. Even yet, the spatial relationship could have been closer overall. Only 22 city pairings had strong connections in 2011, making up 7.3% of the total (see the section below for the first 5% of the sample data). The cities with the highest connection intensities were the top cities, Shanghai and Suzhou, and individual province capital cities in the middle and lower reaches. A total of 118 city pairings, or 39.3% of the total, had weak connection intensities. There were 55 city pairs with a vigorous link intensity by 2021, making up 18.3% of the total. A total of 73 city pairings, or 24.5% of the total, had weak connection intensities. This demonstrates how the state strengthened its financial and talent support for local technological innovation as metropolitan agglomerations' economic development matured. The center cities dominated by urban areas saw a considerable improvement in the efficiency of industrial-green-technology innovation, strengthening the correlation between cities.

　　Subregionally, the Yangtze River Economic Belt's industrial-green-technology innovation-efficiency spatial correlation exhibited clear signs of spatial disequilibrium and displayed a three-plate pattern in the geographical structure. Specifically, a comparatively dense spatial-connectivity network developed between the cities in the lower parts of the Yangtze River Economic Belt. The Yangtze River Delta region lies in the center of China's "T" growth strategy. Market expansion, industrial labor division, and regional traffic accessibility were flawless. The urban spatial-connection network had the most extensive network of spatial connections. The geographic flow of production elements, including people, logistics, and information flow, was frequent. A spatial network centered on provincial capital cities was steadily developed in the middle reaches of the Yangtze River Economic Belt. However, the urban network system could have been flawless. The central performance was the dominance of major cities like Wuhan, Nanchang, and others; this lack of diffusion impact caused the outlying areas to develop slowly. There were few connections between each other. The Yangtze River Economic Belt's upstream region constructed a spatial-link channel with Chongqing and Chengdu at its center. Yet, it is still being determined whether provincial capital cities like Kunming and Guiyang had polarizing effects. Moreover, due to the distance-attenuation law in spatial interaction, there was no spatial-link network between the upstream cities. The economic ties between each city and Chongqing were the main focus of the spatial interaction. There was generally little spatial contact between cities' middle and lower reaches and other cities.

### 3.2. Whole-Network Analysis

Figure 2 illustrates the increasingly complicated network structure of industrial-green-technology innovation in the Yangtze River Economic Belt. Cities not near one another overcame the limitations of geography, created cross-regional connection effects, and progressively developed the spatial-structure characteristics of polar diffusion. The downstream area had a substantially higher degree of spatial correlation than the upstream area. In particular, there were 1207 network relationships in 2011, and the network-intensive regions were mainly confined to the agglomerations downriver. Particularly in the top reaches of Yunnan, Guizhou, and other places, the spatial link between the middle and upper reaches could have been more varied. This reflects that the inter-provincial linkages were artificially broken, there was no long-term division of labor and cooperation across regions, and the pervasive local protectionism resulted in the construction of a vassal economy, resulting in impediments to inter-regional development. Administrative orders compelled a few temporary task divisions. The total number of networks that would be connected by 2021 was 1861. The downstream areas were more tightly correlated with other regions due to the abundant innovation resources and high level of innovation. The connections between cities in the middle and the upstream regions became more potent due to the active application of Western development strategy and the expansion of central China. On the other hand, as the number of central cities expanded over the study period, a multi-center-driven network structure eventually developed in the spatial correlation network. The core–periphery structure of the technical-innovation-association network was visible. At the network center, one tendency was strong and strong. Historically, the association network centered on Chongqing, Shanghai, Chengdu, and Suzhou, each crucial to the network. Most of the cities in the Yangtze River's upper reaches are situated on the network's shore. However, further application of the network-analysis method for quantitative analysis was still required for the overall network characteristics and the specific roles each network node played.

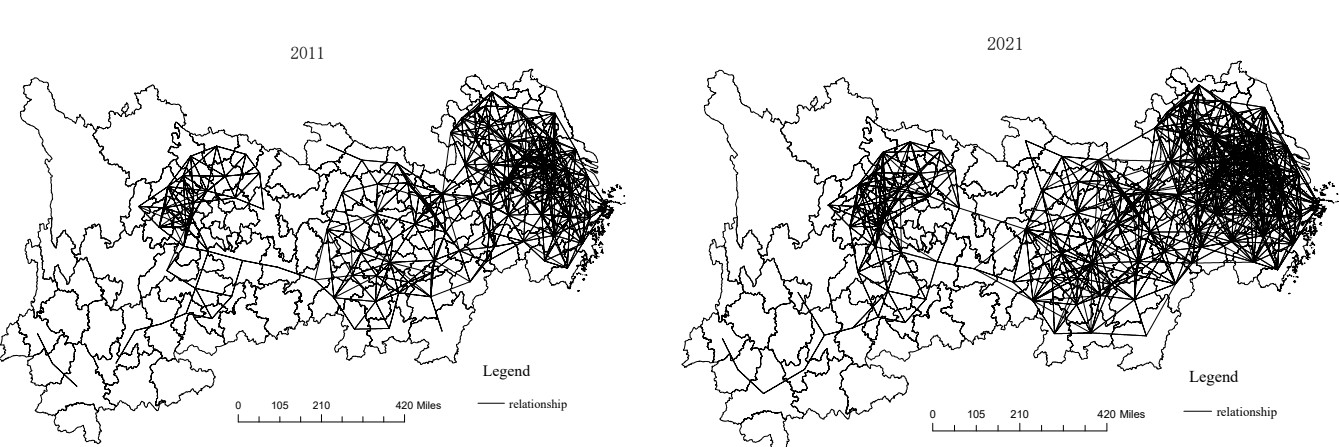

**Figure 2.** Spatial-correlation network of industrial-green-technology innovation in the Yangtze River Economic Belt.

To further characterize the overall structural characteristics of the spatial-correlation network each year, this paper used Ucinet software to measure the structural characteristics of the network each year, such as the number of relationships, network density, network correlation, network efficiency, and network hierarchy, as shown in Figures 3 and 4.

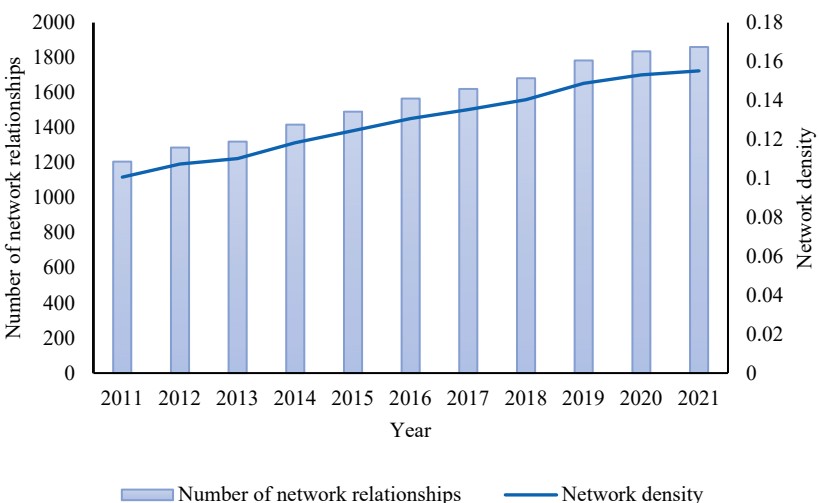

**Figure 3.** Spatial-association network density and network relationship.

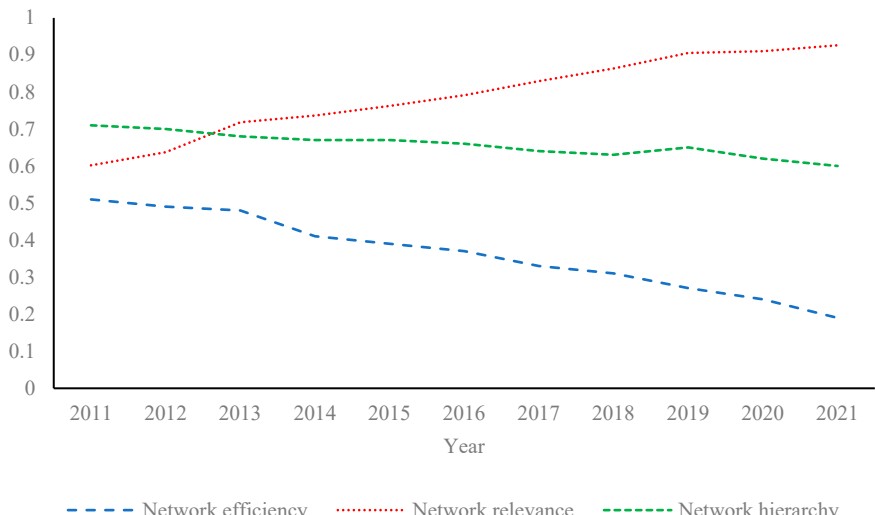

**Figure 4.** Spatial-correlation network efficiency, network rank, and network correlation.

Figure 3 illustrates the growing trend in the Yangtze River Economic Belt's network density, which rose steadily from 0.10 in 2011 to 0.16 in 2021. This was primarily due to the state's promotion of objectives or policies like the innovation-driven development strategy and high-quality development, which sped up the cross-regional flow of innovative resources like knowledge, technology, and talent and caused the innovation linkage between various regions to increase continuously. Yet, it was still low compared to the theoretical maximum achievable density of 1, showing that there is still much space for development in the association between the efficiency of industrial-green-technology innovation and cities. It is necessary to enhance city-to-city innovation cooperation further. The more redundant connection lines were added along with an increase in value. Therefore, the network density was sometimes better when it was more significant. The interactive cost of green-innovation elements between regions will rise until it reaches the network's tolerance threshold, and the spillover and buildup of features will be prevented. Thus, network efficiency and hierarchy analysis must be used to determine whether the spatial-correlation network is good.

The fewer connections, the simpler the network and the more unstable the structure, the greater the network efficiency. The network efficiency of industrial-green-technology innovation in the Yangtze River Economic Belt fluctuated downward during the study period, from 0.51 in 2011 to 0.19 in 2021, indicating that the spatial-network structure

stabilized and the correlation channels of industrial-green-technology innovation efficiency among cities gradually increased. The spatial-correlation network's hierarchical properties eroded progressively, as shown by the downward trend at the network level. Yet, the total network level ranged from 0.6 to 0.71, indicating a higher industrial-green-technology innovation rate in the Yangtze River Economic Belt. Cities' industrial-green-technology innovation-efficiency levels exhibited clear hierarchical patterns and a certain amount of solidification. The network had a clear core–edge structure, which suggests that cities play different roles in the spatial correlation network and that the network structure has to be further improved. In addition, the network's overall accessibility was greatly enhanced, with the network-correlation degree rising from 0.60 in 2011 to 0.926 in 2021. Cities' adoption of industrial green technologies was generally associated. Unfortunately, the network-correlation degree did not reach 1 since some sections of the upper Yangtze River did not build linkages with other cities.

According to an analysis of the characteristics of the overall associated network structure's evolution, in the early stages of the Yangtze River Economic Belt's development, the disparities in institutional policies and development patterns to some extent deterred cross-regional cooperation and communication and impeded the distribution of resources for green innovation. The Yangtze River's provinces and cities have low levels of communication and collaboration, which is reflected in the low number and density of network links. A further barrier to the flow and interoperability of interim information and technology is the rigorous network level and high network efficiency. The Yangtze River Economic Belt strategy is being developed further, which has accelerated marketization and basin integration while gradually erasing the boundaries of the interval system. The fundamental role of the market mechanism in resource allocation has become increasingly important, which has stimulated regional green innovation to some extent; accelerated the flow of resources like capital, technology, and talent among different regions and industries; and gradually dismantled the strict hierarchical spatial structure of the regional industry's efficiency of green-technology innovation. The Yangtze River Economic Belt's ideal distribution of industrial-green-technology innovation space depends on improving regional coordination and cooperation, modifying the internal organization of the river-basin network, appropriately increasing network-correlation density, and maintaining the stability of the overall network structure.

*3.3. Subsection*

By measuring the degree centrality, intermediate centrality, and closeness centrality of industrial-green-technology-innovation spillover-network points in the Yangtze River Economic Belt, we can further explore the status, role, and change trend of cities in the green-technology-innovation spillover network, which is conducive to clarifying the formation mechanism of the industrial-green-technology-innovation spillover network in the Yangtze River Economic Belt. Therefore, based on the binary matrix, this paper used the multiple-measures algorithms in UCINET software to calculate the point centrality, closeness centrality, and intermediate centrality, which reflect the characteristics of network nodes. Taking the average value as the dividing line, the higher average value was divided into the medium–high and high levels, and the lower average value was divided into low and medium–low levels. ArcGIS software was used for visual display, as shown in Figures 5–7.

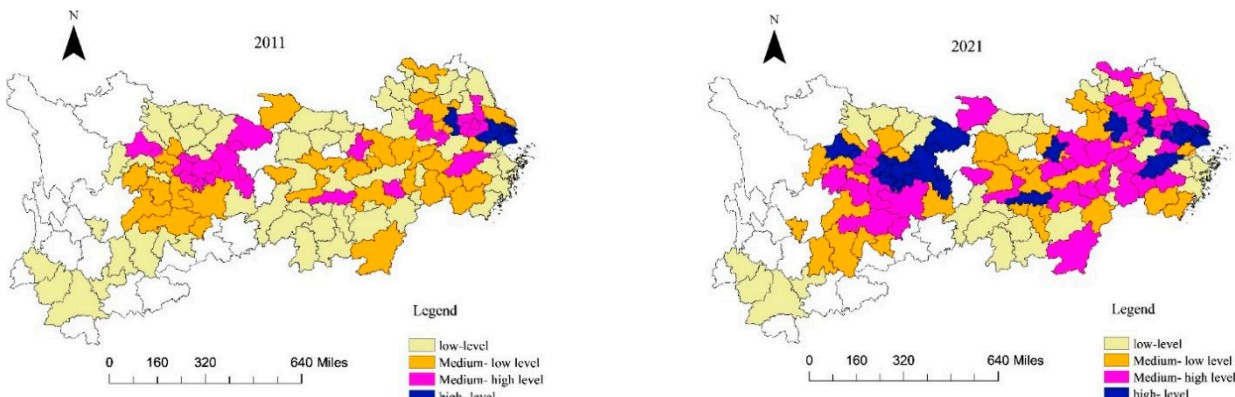

**Figure 5.** Degree centrality of industrial-green-technological innovation efficiency in the Yangtze River Economic Belt.

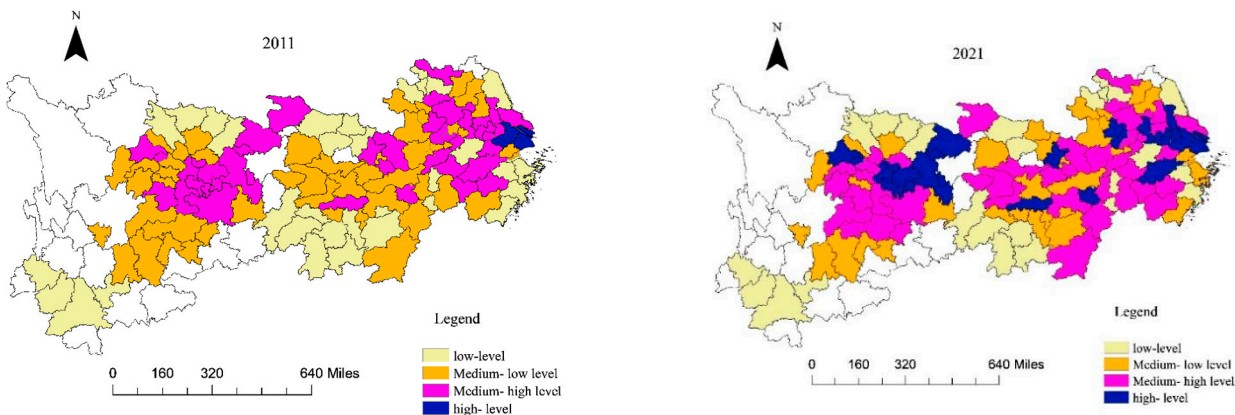

**Figure 6.** Closeness to centrality of industrial-green-technology innovation efficiency in the Yangtze River Economic Belt.

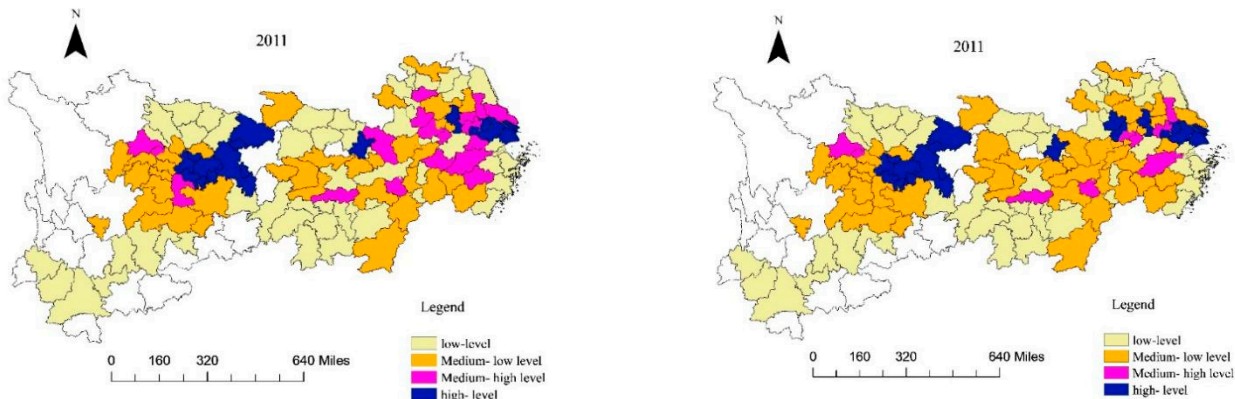

**Figure 7.** The betweenness centrality of industrial-green-technology innovation efficiency in the Yangtze River Economic Belt.

### 3.3.1. Degree Centrality

First, the control of one network region over another was more robust the higher the point centrality. During the study period, there was an increase in the Yangtze River Economic Belt's industrial-green-technology innovation efficiency. In 2011, the degree centrality had an average value of 18, with greater importance in 36 and 63 cities. By 2021, the degree of centrality had an average value of 25.22, whereas 74 cities, mainly in the central and western areas, had discounts more significant than the average. Second, the degree of centrality was high in the Yangtze River Delta, Shanghai, Nanjing, Hangzhou,

Suzhou, and other locations, indicating that the metropolis had a powerful radiation capacity over the entire network. The ties between other regions were more robust, and the midwestern region is where the siphon effect was most noticeable. Chongqing, Chengdu, Wuhan, and other cities were more apparent in the middle and upper parts of the region. The point center greatly aided the connection, which was higher than the other western regions and had the fastest growth. The southwest border cities of Guizhou and Yunnan were on the fringe of the entire network and had limited interaction with neighboring cities. However, as the economy has grown, particularly with the Belt and Road Initiative and the national Yangtze River Economic Belt plan strengthening, more provinces and cities are now part of the green-innovation network. The degree of centrality has increased.

### 3.3.2. Closeness Centrality

Regarding closeness centrality, a region is said to have a high value if it can be connected to many other areas over a reasonably short journey. One part will likely impact adjacent regions at a distance from the network's core. Figure 6 shows that, in the early stages of the study, there were no appreciable differences in the proximity centrality of industrial-green-technology innovation efficiency among the cities in the Yangtze River Economic Belt. Only Shanghai, Suzhou, and Wuxi were at a high level of proximity centrality, suggesting that these cities have always been key players in the spatial-correlation network and are more likely to be directly connected to other cities, supporting the efficiency of green innovation in other cities' tourism sectors. By 2021, the Yangtze River Economic Belt's closeness centrality to industrial-green-technology innovation efficiency had substantially enhanced. Cities at a higher level were primarily concentrated in the Yangtze River Delta, Wuhan, Chengdu, and other cities, showing that the Yangtze River Economic Belt's industrial-green-technology innovation-efficiency network was becoming flat and that the spatial network's correlation and flow efficiency were improving.

### 3.3.3. Betweenness Centrality

The number of shortcuts in many regions increased along with the betweenness-centrality value, which also affected how prominent the bridge and intermediary roles were in the network. The analysis's findings indicate that the basin had three central intermediary nodes. The following are the justifications for placing Chengdu in the upstream region, Wuhan in the middle, and Shanghai in the downstream part. The central city of the metropolitan area occupied significant information-transmission channels with the continuous improvement in economic-development level and the continuous optimization of the innovation environment due to its strong influence on the economy, culture, and other aspects. It functioned as an intermediary and bridge in the geographical-correlation network and significantly impacted the correlation of other cities' green-innovation efficiency. It was a critical node in the system and was firmly in charge of how the connections between the various provinces were made. Each central city paid attention to the development of green-technology innovation and implemented an innovation-driven development plan led by significant data intelligence, which strengthened the investment of innovation resources and the transformation of scientific and technological achievements. This was especially true with implementing the Central Rise Plan and the Western Development Plan. Transferring industrial technologies, environmental governance, cooperative defense, and command was essential. Traffic, infrastructure, financial support, and the siphon effect, which had relatively little control over the green-innovation spillover networks and was in a passive communication position, may all have contributed to the cities in the Yunnan–Guizhou region having low intermediary centrality.

To sum up, the Matthew-effect characteristics of urban green-innovation efficiency in the Yangtze River Economic Belt were apparent: the downstream cities could effectively transform various green-innovation resource factors, and the development level of green innovation was relatively high, which was in a fairly central position in the spatial network of urban green-innovation efficiency in the Yangtze River Economic Belt; affected by

geographical conditions, urban-development level, and economic level, the attraction of upstream and middle cities to green-innovation resources was relatively weak, often in a fairly passive position in the spatial network, and could not form a positive interaction with other cities.

3.3.4. Block-Model Analysis

From the above analysis, it can be seen that the status and role of each province in the network were heterogeneous and regional differences were apparent. To further reveal the role of each region in the network and describe the interactive relationship between regions, based on the correlation network of green-technology innovation efficiency in 2021, 110 prefecture-level cities in the Yangtze River Economic Belt were divided into four plates using the Ucinet-software CONCOR algorithm and selecting a maximum segmentation depth of 2 and a concentration standard of 0.2, as shown in Table 3.

**Table 3.** Composition of block members.

| Block Name | Number of Cities | Cities Included |
|---|---|---|
| I (bi-overflow) | 11 | Huzhou, Huai'an, Kunming, Ganzhou, Nanchang, Guiyang, Wuhu, Jinhua, Zhenjiang, Yangzhou, Lishui |
| II (agent) | 53 | Lianyungang, Changde, Chuzhou, Suqian, Lu'an, Bengbu, Huainan, Maanshan, Yibin, Anqing, Xiangtan, Fuyang, Yueyang, Jiaxing, Zunyi, Yongzhou, Luzhou, Quzhou, Shangrao, Bozhou, Suzhou, Huangshi, Leshan, Jingzhou, Yichang, Shiyan, Jingmen, Xiaogan, Huanggang, Nanchong, Xiangyang, Zhuzhou, Dazhou, Hengyang, Shaoyang, Deyang, Mianyang, Jiujiang, Suining, Chenzhou, Yichun, Huaihua, Ji'an, Xuancheng Zigong, Neijiang, Loudi, Xianning, Guang'an, Taizhou, Fuzhou, Jingdezhen, Ezhou |
| III (net overflow) | 18 | Shanghai, Nanjing, Wuxi, Xuzhou, Suzhou, Nantong, Wenzhou, Taizhou, Yancheng, Changsha, Chengdu, Wuhan, Hangzhou, Hefei, Shaoxing, Ningbo, Chongqing, Changzhou |
| IV (net benefit) | 28 | Pu'er, Zhangjiajie, Yingtan, Lincang, Lijiang, Baoshan, Pingxiang, Zhoushan, Panzhihua, Chizhou, Suizhou, Guangyuan, Huangshan, Meishan, Bazhong, Anshun, Tongling, Yuxi, Ya'an, Xinyu, Huaibei, Ziyang, Yiyang, Liupanshui, Bijie, Tongren, Qujing, Shaotong |

From the perspective of spatial distribution, the 18 cities of plate III included 14 downstream cities, 2 midstream cities, and 2 upstream cities. The members of plate III had mainly developed cities in the lower reaches of the Yangtze River and some provincial capital cities in the upper and middle reaches. These cities' scientific-research institutes, high-tech enterprises, and other innovation subjects were gathered. Innovation factors such as knowledge, technology, and capital were abundant. The level of technological development was high, and the technical-innovation ability was strong. The 11 cities of plate I included 7 downstream cities, 2 central cities, and 2 upstream cities. In addition, 14 of the 16 cities in Anhui Province were located in plate II and plate IV, indicating that Anhui did not integrate into the technological-innovation correlation network of Shanghai, Zhejiang, and Jiangsu and that there is room for optimization in the downstream-correlation network. Plate II included 53 cities, of which 14 were located downstream, and the upstream cities were located in the middle and upper reaches. Plate 4 comprised 28 cities, mainly Yunnan, Sichuan, and Anhui.

In addition, according to the four plates' internal and external relations, each plate's role or position in the spatial-correlation network of industrial-structure upgrading in the Yangtze River Economic Belt was further revealed, as shown in Table 4.

**Table 4.** Spatial correlation of industrial-green-technology innovation efficiency in the Yangtze River Economic Belt.

| Block | Number of Accepted Relationships | | Number of Issued Relationships | | Block Roles |
|---|---|---|---|---|---|
| | Intra-Block | Out of Block | Intra-Block | Out of Block | |
| I | 1437 | 305 | 1437 | 426 | Two-way spillover |
| II | 252 | 354 | 252 | 313 | Broker |
| III | 348 | 302 | 348 | 284 | Net overflow |
| IV | 401 | 115 | 401 | 84 | Net benefit |

Table 4 shows that the first plate had 1863 spillover relations, 1742 acceptance relations, 426 spillover relations, and 305 acceptance relations. The number of overflow relationships outside the plate was substantially higher than the number of acceptance links, and the plate had many acceptance and spillover relationships. As a result, the plate can be placed in a two-way exit position, showing that the lower reaches of the Yangtze River had a higher level of industrial-green-technology innovation that created a linkage-innovation effect with other regions. There were 565 total relationships issued by the second plate, 313 of which were sent to the plate, 606 entire acceptance relationships, and 354 absolute external relationships. The number of relations within the plate was lower than 252, and the acceptance and spillover relations were more prevalent. As a result, the plate can be described as a broker position, indicating that the Yangtze River Economic Belt cities' efficiency in terms of green innovation allowed them to transfer and allocate different green-innovation resources and elements in a reasonable manner as well as act as a bridge and betweenness in the network-spillover path. The third plate had 610 acceptance relations and 683 spillover relations, of which 262 and 345 were outside the plate, respectively. To be referred to as a net-spillover posture, more spillover relations must exist than acceptance relations. The fourth plate featured 416 total acceptance relations, 485 total spillover relations, and only 115 and 84 total relations outside the plate, respectively, showing that the plate had limited interaction with the outside environment. The plate's interior contained most of the correlation relations, allowing it to be placed in a location that promoted net spillover. The provinces and cities gradually transitioned from discrete to joint, breaking the traditional administrative boundary limit, as shown. Diffusion radiation caused the clustering of the basin's cities and regions. The Yangtze River Economic Belt's geographical-correlation network of green-innovation efficiency was rapidly becoming more integrated.

To more clearly reflect the spillover effect and transmission path between the plates, it was necessary to calculate the density matrix of each plate according to the distribution of the correlation between the plates. In 2021, the global network density was 0.16. If the lattice value in the plate-density matrix was greater than the worldwide network density, it was assigned to 1 in the matrix and vice versa. The calculation results are shown in Table 5. Finally, the density matrix was converted to an image matrix. The interaction between the four plates based on the image matrix is shown in Figure 8.

**Table 5.** Density matrix and image matrix of industrial-green-technology innovation efficiency in the Yangtze River Economic Belt.

| Block | Density Matrix | | | | Image Matrix | | | |
|---|---|---|---|---|---|---|---|---|
| | I | II | III | IV | I | II | III | IV |
| I | 0.23 | 0.18 | 0.25 | 0.04 | 1 | 1 | 1 | 0 |
| II | 0.17 | 0.20 | 0.31 | 0.12 | 1 | 1 | 1 | 0 |
| III | 0.16 | 0.31 | 0.24 | 0.16 | 1 | 1 | 1 | 1 |
| IV | 0.11 | 0.17 | 0.03 | 0.18 | 0 | 1 | 0 | 1 |

Note: "1" means row-pointing column, "0" means no correlation.

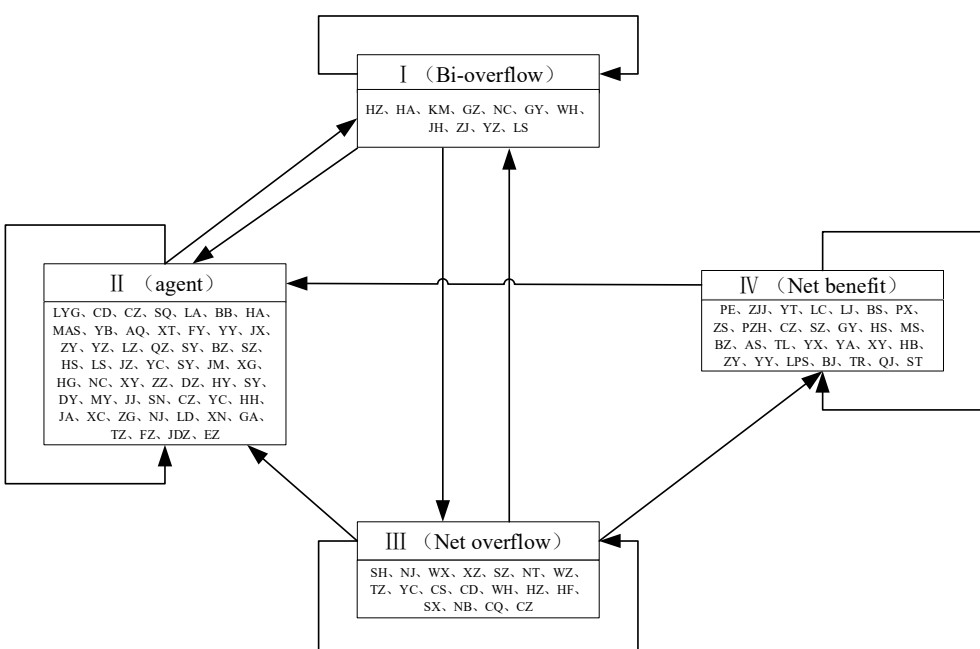

**Figure 8.** Interaction diagram of four significant blocks.

The correlation between plates I, II, and III was close, and there was a connection between the two plates in terms of technological progress. Contrarily, plate IV only created a technical relationship with plate I by absorbing plate III's technological spillover. The fact that plate III exhibited an internal correlation of technical innovation and generated technological spillover to plates I, II, and IV shows how important it was for the developed cities in the lower reaches to be able to spread technological innovation, making them the engine of technological-innovation growth in the Yangtze River Economic Belt. Plate II's internal connection was fragile. It was, however, intimately tied to plates III and I, which served as brokers, encouraged the dissemination of technological-innovation components and accomplishments, and offered crucial support for the spatially coordinated growth of technological innovation in the Yangtze River Economic Belt. The fact that the plate IV members were situated at the network's edge may be related to the lower-than-average economic development of the underdeveloped cities in the upper and middle reaches, insufficient technical infrastructure, a lack of innovation-related motivation, and a limited capacity for technological innovation. Only plate III's spillover in technical innovation was received.

The Yangtze River Economic Belt's spatial-correlation network-transmission mechanism for green-innovation effectiveness had noticeable gradient-spillover characteristics. The Matthew effect demonstrated that although backward places are more likely to produce a reverse spillover of innovation elements, more developed regions are more likely to absorb green-innovation aspects. From a spillover-path perspective, we should encourage the linkage of green-innovation efficiency between provinces and cities in the basin to a higher level of chain spatial-network stages, strengthen regional collaboration and linkage capacity, widen the channels for innovation interaction and communication, and concentrate on resolving the issue of development-path dependence and transmission-mechanism locking of technological innovation in the central and western regions. To address the issue of a homogeneous product, each province and city should identify the significant direction of its dislocation development in the high-quality development of the Yangtze River Economic Belt in conjunction with location circumstances, resource endowment, and economic foundation.

## 4. Conclusions and Suggestions

This study used the Super-SBM model based on unexpected output to measure the industrial-green-technology innovation efficiency of 110 cities in the Yangtze River Economic Corridor from 2011 to 2021. Based on the modified gravity model, a geographical-correlation network of industrial-green-innovation effectiveness in the Yangtze River Economic Belt was built. Social-network analysis examined the overall network properties and node centrality of the green-innovation-spillover network. The flow direction of green-technology innovation among cities and the function of cities in the network of green-innovation spillover were clarified. At the same time, the block model was employed to categorize cities with the same function status, and the network density between various plates was used to clarify the relationship between regions. The specific findings of the study are as follows:

The main conclusions are as follows. (1) Overall, during the study period, the connection intensity of industrial-green-technology innovation efficiency among different cities in the Yangtze River Economic Belt was mainly dominated by leading cities Shanghai and Suzhou and individual provincial capital cities in the middle–lower reaches. (2) During the study period, the network density of the Yangtze River Economic Belt showed an upward trend, the network efficiency showed a fluctuating downward trend, and the spatial-network structure tended to be stable, with a more prominent core–edge structure. (3) During the study period, the degree of centrality of industrial-green-technology innovation efficiency in the Yangtze River Economic Belt showed an upward trend. There was little difference in closeness centrality. Regarding betweenness centrality, there were always three core intermediary nodes in the basin: Shanghai in the downstream area, Wuhan in the middle reaches, and Chengdu in the upstream area, which played the roles of intermediary and bridge in the spatial-correlation network. (4) In the block-model analysis results, the spatial relationship between two-way spillover, broker, and net spillover was relatively close. There was a technological-innovation relationship between any two plates. The net-overflow section only accepted the relationship from the net-overflow section and issued the relationship to the broker section. Overall, the spatial correlation of green-innovation efficiency of cities broke the traditional geographical spatial restrictions, showing a more complex and stable spatial relationship.

The following policy recommendations are based on the linking results and regionally coordinated development of industrial-green-technology innovation in the Yangtze River Economic Belt:

(1) Advance the regionally coordinated growth of industrial-green-technology innovation in the Yangtze River Economic Belt and fully comprehend the peculiarities of spatial-network structure. There is a significant spatial association between the effectiveness of industrial-green-technology innovation in the Yangtze River Economic Belt. Each city must work to increase technological-innovation efficiency, but it also depends on other cities. To achieve the coordinated development of various sectors, it is necessary to define the position relationship and role orientation of each town, pay attention to the complex relationship between cities from the perspective of space, formulate pertinent policies to promote industrial-green-technology innovation and high-quality development, and promote cross-regional and cross-sectoral collaboration of innovation resources and factors through macro-control and market mechanisms.

(2) Make full use of the impact of spatial-network structure and encourage the growth of innovative urban industrial green technology. Actively look for ways to increase the geographical correlation of industrial-green-technology-innovation effectiveness and continue to use the spatial-network structure to help the growth of green-technology innovation in each city and the entire area. Increase the interaction between cities, the stability of spatial-correlation networks, and the growth of regional green-technology innovation by expanding communication channels between cities in industry, market, information, and other regions. This improves the discourse power of subordinate cities and effectively promotes the coordinated and balanced development of green-technology innovation across

various cities. This is done by strengthening the feedback of cities with higher efficiency of green-technology innovation to lower cities, reducing the gap between cities in terms of innovation elements, customer market, human capital, capital, and other aspects.

(3) The Matthew effect of the Yangtze River Economic Belt's green-innovation efficiency is broken, according to the features of each city node and each plate in the network for green innovation. For each city in the network of green-innovation spaces to form a harmonious interaction, exchange, and balance of various green-innovation resources and factors, the downstream cities fully exploit their advantages. This helps to drive the green-innovation development of the middle and upper reaches of the cities and optimize the overall pattern of the green-innovation efficiency of the cities in the Yangtze River Economic Belt. They gain an understanding of the internal relationships between the growth of green-innovation efficiency between cities based on their placement in the associated network, taking the expansion of spatial-spillover channels as a critical indicator, placing their development on the chessboard of the basin's overall development, determining the direction and path of dislocation development, and realizing local and global coordination.

(4) To promote green growth, we integrated regional differentiation strategies. As the center of green-technology innovation in the Yangtze River Economic Belt, we should be mindful of the relationships between and the spillover effects of each plate in the correlation network. We should also actively direct and drive the intermediary plate and the net-spillover plate to improve the innovation capacity and technology of green technology while continuously raising the level of green technology. The Yangtze River Economic Belt's cities must gradually increase the correlation density and innovation efficiency of green-technology innovation by increasing the active participation and learning consciousness of intermediaries and net-spillover plates, taking full advantage of their advantages, introducing and absorbing advanced green-development technologies, and strengthening exchange and cooperation among regions.

The current research sample of this paper was 110 prefecture-level cities in the Yangtze River Economic Belt from 2011 to 2020, but some cities had missing data, such as Bijie and Tongren. Therefore, the interpolation method was selected to calculate, which was not accurate. In addition, different scholars have different methods and even indicators for calculating the efficiency level of green-technology innovation. The final results will inevitably have a certain degree of gap. In spatial-correlation-strength analysis, due to a large number of research samples, only the data with a spatial-correlation strength in the top 5% were selected for display, leading to a lack of description of the overall sample. At the same time, the study of the spatial-correlation network of industrial-green-technology innovation was not compared with global green-technology innovation, which makes the research lack a global–local scale-analysis perspective. The Yangtze River Economic Belt spans the three central eastern, middle, and western regions. The differences in location make cities have different natural conditions and resource endowments. The stages of urban development, industrial-structure characteristics, and goals were different, and the emphasis on green development was also different. Therefore, cities can be classified according to different standards and analyzed for their impact mechanism. In addition, the research shows that the spatial-correlation network of green economic efficiency had a relatively strict hierarchical structure, and the network structure needs to be further optimized. Based on the evolution trajectory of spatial-correlation networks and the influencing factors of spatial heterogeneity, exploring the spatial-optimization model of future network structure is also an important direction worthy of exploring further.

**Author Contributions:** Conceptualization: M.Y.; methodology: M.Y.; software: M.Y.; validation: M.Y., Z.L. and Y.W.; data curation: Z.L. and Y.W.; supervision: Z.L. and Y.W.; All authors have read and agreed to the published version of the manuscript.

**Funding:** 2022 Huzhou Normal University talent research start-up funds, grant number RK19075.

**Institutional Review Board Statement:** Not applicable.

**Informed Consent Statement:** Not applicable.

**Data Availability Statement:** The required panel data mainly come from the China Statistical Yearbook, China Environment Statistical Yearbook, China Energy Statistical Yearbook, and China Science and Technology Statistical Yearbook, and the patent authorization in the output index is expected to come from the patent-retrieval system of the State Intellectual Property Office of the People's Republic of China.

**Conflicts of Interest:** The authors declare no conflict of interest.

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
