# Peer review of "Features of Industrial Green Technology Innovation in the Yangtze River Economic Belt of China Based on Spatial Correlation Network"

_sustainability, doi:10.3390/su15076033_

Round 1
Reviewer 1 Report
This article by Mengchao Yao and Jinjun Duan entitled “Features of Industrial Green Technology Innovation in the Yangtze River Economic Belt Based on Spatial Correlation Net- work”. I have gone through the manuscript and it is an interesting work. This research work can improve industrial green technological innovation in the Yangtze River Economic Belt. Overall, the whole manuscript is well written, the study is well designed and executed, data properly analysed, methodology detailed, and results well discussed.
1. All figure legends should be revised and should write in details.
2. Conclusion and suggestions is very large. The readers will feel boring and also feel confusion. The authors just mentioned the methodology and results. I suggest to the authors that they should rewrite conclusion precisely according to title and objectives.
Author Response
请参阅附件。

Reviewer 2 Report
The authors present a paper entitled “Features of Industrial Green Technology Innovation in the Yangtze River Economic Belt Based on Spatial Correlation Network” in which they analyze the evolution characteristics and operating mechanism of the spatial correlation network structure of green technology innovation efficiency based on "relational data" from the perspective of a complex network in order to enhance the collaboration and effective allocation of innovation resources and optimize the spatial distribution of green industrial technology innovation.
The subject of the article is very specific, but interesting, and falls within the scope of the journal. I recommend a minor revision of this manuscript because some improvements should be made before publication:
Title:
The authors should add the word “China” in the title, since “Yangtze River” is a very specific site not necessarily known by readers outside China.
Introduction:
The authors should explain in more detail in the introduction what is new about the work and why this study is necessary. Maybe it is somewhere in the last paragraph of the introduction (Lines 119 to 137), but this section is very hard to read. It should therefore be rewritten more clearly.
Material and methods:
In line 193 the authors include “sulfur dioxide (SO2) emissions”. Why these emissions specifically? Why no others? In general, the authors should explain better why they included these inputs.
Conclusions:
The conclusions are very long and should be reduced to one or two paragraphs including the main highlights of the study.
References:
The authors need to check the references because the numbers are doubled (1. [1] Du K…)
Regarding tables and figures, the maps in figure 1 are hard to see, they should be enlarged. Moreover, figure 8 quality must be improved.
Finally, although there are no critical errors regarding the English language, the manuscript should be reread carefully by the authors and all paragraphs should be checked because many sentences are strangely isolated or do not make sense, making it difficult to follow the manuscript as a reader (For example in line 123, after a dot: “The policies and conclusions reached the urban level are more indicative”, or line 657: Clarified is the flow direction…”).
Reviewer 3 Report
The title of the manuscript reflects the purpose and contents of this manuscript well. The abstract and conclusion section contains comprehensive studies’ highlights. Significantly, the author well organized the meaning of each result for the discussions in the Conclusion part.
It would look better if the authors could modify some parts of this manuscript.
First, the tables could look better readable. So, please organize and rearrangement of the format for the readers.
Second, please add the horizontal and vertical tittle and units of the graph in figure 3 and 4. It could be helpful for the readers to understand the results just through the figures.
Third, table 2 in 3.3.4 Block model analysis results needs to be moved to the methodology part, and please describe how the authors set up the data and modeling parameters for the block model in 2.3.3. part of the methodology.
Thank you for contributing the sustainability journal.
Reviewer 4 Report
The current manuscript can be published in Journal of Sustainability after revision as mentioned below:
1. In the last paragraph of the introduction section, the author should write the research objectives clearly.
2. Table 1. is presented on a separate page, it should be on the same page.
3. To make it easier for the reader, the author must draw Figure 1-7 more precisely with contrasting and more communicative colors..
4. The discussion section needs to be more straightforward and could be more detailed. The focus is also here on the additional value that this article provides regarding the current literature in comparison with the findings of the earlier literature. The future aspect also needs to be mentioned in this section..
5. The conclusions of this study are too long; the author should be written a more concise.
